# Experimental Study on the Coating Removing Characteristics of High-Pressure Water Jet by Micro Jet Flow

**DOI:** 10.3390/mi12020173

**Published:** 2021-02-10

**Authors:** Dayong Ning, Qibo Wang, Jinxin Tian, Yongjun Gong, Hongwei Du, Shengtao Chen, Jiaoyi Hou

**Affiliations:** 1National Center for International Research of Subsea Engineering Technology and Equipment, Dalian Maritime University, Dalian 116026, China; ningdayong@dlmu.edu.cn (D.N.); wqb201908@dlmu.edu.cn (Q.W.); wfg201308@gmail.com (J.T.); gyj@dlmu.edu.cn (Y.G.); nicholasdo@dlmu.edu.cn (H.D.); dmucst@dlmu.edu.cn (S.C.); 2The State Key Laboratory of Fluid Power and Mechatronic Systems, Zhejiang University, Hangzhou 310027, China

**Keywords:** water jet, coating-removal characteristics, micro jet impingement, rotating cleaning disc

## Abstract

In this paper, coating removal characteristics of water jet by micro jet flow affected by cleaning parameters is analyzed. Numerical simulation of fluid field calculates the velocity and pressure distribution of a water jet impinging on a rigid wall, which is used for design experiments of coating removal affected by jet pressure, traversal speed, and repeated impacting times. The removal width is used as a measure of water jet coating removal capability. Experiment results show that the coating removal width is constant, independent with traversal speed or repeated times when total exposure time of waterjet impingement is fixed. According to results of coating removal by a linear moving water jet, this study also analyzes characteristics of coating removal by rotating jet disc, especially residual coating affected by rotational and moving speed of the cleaning disc. The research is helpful to improve the coating removal efficiency of cleaning disc devices.

## 1. Introduction

Remanufacturing cleaning technology is a method of cleaning and removing complex dirt adhering on surface of waste material using devices or chemical solvent. The cleaned product units can meet the requirements of cleanliness of analysis, detection, fix, processing, and so on. The technique of coating and rust cleaning on the steel structure surface is mainly about peening, chemical corrosion, and water jet cleaning. High pressure water flowing through the micro flow channel of the nozzle generates a high fluid-velocity water jet. When water jet impacts the painted plate, the kinetic energy of micro jet flow convert into pressure energy causing coating failure. Compared with the other two cleaning methods, water jet cleaning has characteristics of good security, non-pollution, recovery of impurities, high adaptability, and low cost. The surface cleaned by the water jet has a high degree of smoothness and little salinity which is fast-drying and has hardly any rusting. Water jet cleaning technology is widely used in the cleaning of vessel outer walls, aircraft skins, oil storage tanks, heat transfer equipment, and so on [1]. The parameters and applications of the existing water jet equipment are shown in Table 1.

Studies on water jet cleaning are mainly about cleaning efficiency, different material removal mechanism by water jet impingement, and structural design of cleaning equipment. In aspects of numerical simulation and experiments of surface treatment by water jet, Kawale and Chandramohan [2] used CFD simulation to discuss the static pressure and shear stress distribution of jet impingement on the plane with different Reynolds number. The simulation results showed that static pressure and wall shear stress is increasing with Reynolds number. Jet velocity at micro flow channel exit of nozzle must be at least 3 m/s for effective cleaning flat plane. The simulation results were verified by experiments. Chillman et al. [3] used the ultra-high pressure water jet to remove α case of super plastic titanium alloy without damaging the substrate. The effectiveness of jet removal of titanium alloy case was evaluated by experimental method and surface micro-morphology analysis. Liu et al. [4] analyzed the jet velocity turbulent kinetic and void fraction of jet pressure ranging from 80 to 120 MPa. Simulation results showed pressure and velocity cross-sectional distribution of different target distance. This study took comparison of impinged samples by submerged and non-submerged water jet and makes assessment of surface morphology, micro hardness, and surface roughness by experimental method. The study showed that impact force of submerged jet is from jet kinetic energy and cavitation, while the impact force of non-submerged jet converts from jet kinetic energy. Guha et al. [5] took an experimental and numerical study of influence on stagnation pressure attenuation and pressure distribution on the substrate under different target distance, and obtained conclusion that there is a linear relationship between stagnation pressure attenuation and target distance on the jet axis. The optical stand-off distance is 5 times the diameter of nozzle and the jet diffusion radius is 1.68 times the diameter. Zhang et al. [6] analyzed motion characteristics of abrasive particles affected by jet parameters. According to simulation results, it was proved that there is a stagnation pressure zone during jet impingement. An abrasive particle with a large diameter and density has a higher speed and it is hard to change its moving direction. These properties help to improve abrasive waterjet efficiency. Zhang and Chen [7] analyzed coating removal from a passenger-vehicle by waterjet and designed a high-pressure water jet cleaning device for a coating removal rate test. This study showed the effects of jet pressure, moving speed, target distance and inject angle on coating removal rate. And a fitting formula of bumper surface micro coating removal rate was established according to experiment results. Che et al. [8] investigated the process of polishing super hard material surface by abrasive water jet, considering the influence of micro jet impact angle and substrate hardness on surface roughness, and established a mathematical model of a surface abrasive jet polishing super hard material surface roughness. Zhang et al. [9] compared the impact erosion performance of air sandblasting and an abrasive water jet on quartz board surface micromachining. Experimental results showed that an abrasive water jet has brittle and plastic erosion during surface treatment, and surface impinged by an abrasive water jet is smoother than polished by air sandblasting. Terimourian et al. [10] investigated the process of high pressure water jet de-painting organic coating from steel plate surface and the surface micro roughness of steel plate after cleaning by water jet with different target distances and moving velocities. The experimental results showed that coating mass loss increases with jet kinetic energy, and the maximum of mass loss occurs when jet is de-painting with an optical target distance. The substrate topography is not compromised after secondary water jet de-painting. Mabrouki et al. [11] established Ls-dyna 3D finite element model to simulate process of polyurethane coating on aluminum alloy removed by water jet based on momentum conservation and Euler-Lagrangian coupling method. The study analyzed surface stress–strain value with water jet impingement and designed experiment to verify the simulation results of coating surface morphology changed with jet exposure time. Xie and Rittel [12] studied the pure water jet peening process and established a 2D finite element model to simulate a micro jet flow impinging on the target plate. This study calculated jet velocity, pressure, and stress distribution of the plate at a fixed target distance.

In the field of theoretical analysis about jet impingement and coating removal mechanism, Kunaporn et al. [13] established a mathematical model to analyze and calculate the pressure distribution on the surface of aluminum alloy impacted by waterjet at different target distances. The calculation result was used to predict contact pressure and effective peening range. Result of high cycle fatigue test illustrated that surface treatment by micro jet peening can improve fatigue life of aluminum alloy under the optimal target distance. Meng et al. [14] took a study of epoxy-based paint removal by moving water jet. A semi-empirical model was established to calculate coating removal mass by erosion of micro jet droplets based on Springer erosion formula. In order to study the mechanism of jet material removal, Chillman et al. [15] established a mathematical model of jet energy density distribution affected by analyzing the relevant parameters such as micro channel diameter of jet, jet pressure, and traversal velocity. This model was verified by experimental results of titanium alloy trench depth after jet impacting titanium alloy. Mieszala et al. [16] analyzed the erosion mechanism in the abrasive waterjet surface machining process, and carried out the impact test of micro particle with various moving speed. The material removal mechanisms of abrasive jet are different with various crystal structures and surface microstructures. Weiß et al. [17] used jet to separate plastic fiber from textile, and designed jet cleaning device to clean waste carpet and recover plastic components, so as to improve the recovery and utilization of plastic fiber. Hou et al. [18] took theoretical equations to calculate the erosion depth of submerged jets at various target distance and injection angle, and verified the reliability of the model by experimental results of scouring clay. Glover et al. [19] established a mathematical model to calculate the removal width of viscoplastic impurity layer on the smooth surface impacted by fixed and moving jets, and used high-speed photography technology measuring the jet cleaning width at different exposure times to verify the calculation results. Azhari et al. [20] analyzed the three-dimensional surface micro morphology of 304 stainless steel by various pressure jet combination treatments, and found that repeating cleaning with various pressure jets can improve surface smoothness and hardness, as well as prolong the fatigue life of the workpiece.

As for research on cleaning efficiency of the rotating jet and its devices, Peng et al. [21] established a three-dimensional model based on the structure of the cleaning plate on the airport runway jet degumming vehicle to simulate vacuum suction process and the fluid field of internal rotating jet, and optimized operating parameters of the cleaning plate on the basis of simulation result. Borkowski [22] evaluated the cleaning effect of a multi-nozzle device. A theoretical model of surface trace distribution was established, which was used for analyzing cleaning efficiency affected by parameters of trace width, rotation speed, number of nozzles, and traversal speed. Chomka and Chudy [23] took a mathematical analysis of the cleaning trajectory distribution of double-nozzle rotary nozzles, and confirmed the range of the optimal moving speed and angular velocity to make the cleaning trajectory evenly distributed. Momber [24] used image processing method to evaluate the influence of parameters such as micro channel diameter of nozzle, number of nozzles, inject angle, target distance, and flow rate on paint removal effect, and optimized the jet paint removal parameters.

There are relatively few studies on the width of micro jet flow coating removal affected by exposure time, especially the characteristics of jet paint removal on the rough plate. This paper analyzes the process of micro jet flow impinging on the wall with fixed target distance and obtain the theoretical width of impact pressure area. Width of coating removal is investigated under condition of changing traversal velocity, repeated cleaning frequency, and total exposure time. Combined with the application of jet cleaning rotating disc, this study analyzes the removal rate affected by the distribution of jet trajectory, which helps to optimize parameters for improving coating cleanout rate of the rotating cleaning device.

## 2. General Features of Water Jet

### 2.1. Water Jet Structure

The structure of the water jet is shown in Figure 1, which can be divided into three regions: Initial section, main section, and dissipation section with velocity attenuation. When the high-pressure water flows through the micro fluid channel of nozzle, water pressure energy converts into kinetic energy to form a high-speed water jet. Due to dynamic viscosity of fluid and friction of air, the velocity of the micro water jet at the front end will gradually attenuate. The micro flow at the edge of the beam slows down because of air friction and entrainment. Micro flow velocity in this field decreases to zero and the jet beam breaks into micro droplets at the same time. The radius of the jet increases with the target distance. Therefore, the flow velocity distribution at the cross section has the feature that the velocity is high beside the axial line and decreases to zero along the diffusion radius. In the industrial operation, the initial section is used for cutting because of its smaller diffusion radius and high fluid velocity. The main section has a larger diffusion radius and the jet velocity in this region still maintains a high value, so this zone is typically used for cleaning and coating removal operation [12].

### 2.2. Coating Damage Model

In the process of the jet impingement on the coating, former researches show that erosion and surface shearing lead to failure of coating on the ideal smooth surface. When micro jet flow impacts the coating surface, kinetic energy of the jet converts into pressure energy. As impact pressure reaches value of failure strength of the coating, crack grows from the coating surface to the contact plane of substrate during a short time. Water jet impinging on the workpiece breaks the coating and scour trench. Then micro jet flowing along the contact surface impacts the joint of coating and steel plate continuously. Surface shear stress induces micro jet diffusion striping coating from the smooth surface. This phenomenon is typically called “water edge” [19]. The failure principle of coating adhering on the smooth surface is shown in Figure 2.

In the actual coating removal operation, the substrate with surface roughness is used to improve adhesion force between coating and plate. Surface with micro bulges and concaves can change the direction of the jet reflection that the velocity direction of jet micro element is stochastic. So, diffusion jet with various direction hardly forms shear stress along the plate and failure of coating on the rough surface is mainly through jet impingement. Coating removal principle on the rough surface is shown in Figure 3.

The destruction effect of the jet on the coating is mainly generated by the pressure energy converted from the kinetic energy of high speed micro jet flow, so as to overcome the self-strength or adhesion strength to damage the coating [15]. Jet velocity at the nozzle exit and jet diffusion are related to the inlet pressure and nozzle micro flow channel. According to the flow field simulation of the jet nozzle used in the removal experiment of this paper, the diameter of the nozzle micro fluid channel is 1mm, and the inlet pressure is 32 MPa. The fluid field is calculated by *k*-*ε* turbulence model with wall functions. A two-phase flow model is used for calculating water jet impinging on the rigidity wall through the air. The target distance is 10 mm. Simulation result of velocity and surface pressure distributions are shown in Figure 4. Results shows that micro jet velocity is highest at the axis and attenuates to zero along the radial direction at the cross-section. When high pressure water flows through the micro flow channel, pressure energy of water converts into kinetic energy and generates water jet. The energy conversion causes the pressure drop through the nozzle shown in Figure 4c. There is an energy converting region called the stagnant zone around the intersection of the jet axis and the wall, and the reflective jet flows along the wall beyond the stagnant zone [2]. After the jet impacts the wall, the kinetic energy of the jet near the axis is converted into pressure energy, so there is a circular area with a radius of *r* near the jet axis where the workpiece is mainly subjected to the impact pressure. Target distance mainly affects the size of the stagnation pressure area and the maximum impact pressure. In order to study the effect of energy accumulation on micro jet impingement in the process of jet coating removal, further analysis is needed through experiments.

## 3. Experimental Study on the Width of Coating Removal

### 3.1. Experimental Setup

The whole set of the single jet coating removal test system is mainly composed of three parts: High-pressure water generating device, a jet gun, and a ball screw driven by motor. High-pressure water pressurized by the plunger pump is transported through the high-pressure pipeline to the jet gun to form a high-speed jet from the micro flow channel of the nozzle. At the same time, the ball screw drives the jet gun to reciprocate in a straight line to carry out the paint removal test. The schematic and physical diagram of the device and the microstructure size of the nozzle are shown in Figure 5. The traversal velocity regulation range of the ball screw is 0.05 mm/s to 39 mm/s, the maximum stroke is 180 mm, the rated pressure of the jet gun is 38 MPa, and the inner micro fluid channel diameter of the nozzle outlet is 1 mm. The application background of this paper is green coating and rust removal of ship wall. Therefore, the ship steel plate and fluorocarbon paint in hull construction are selected, and the average thickness of coating is 100 μm measured by thickness gauge.

### 3.2. Coating Removal by One Time Jet Impingement

In order to study the effect of traversal velocity on the width of jet paint removal, the experiment of coating removal by water jet with different traversal velocity is designed. In this experiment, the fixed target distance is 10 mm, the jet inlet pressure is 35 MPa, 32 MPa and 30 MPa, the traversal velocity ranges from 0.05 mm/s to 1 mm/s, and the inner diameter of the jet nozzle is 1 mm. The width of paint removal from the area where the jet destroys the coating to the surface of the steel plate is measured after a single traversal of the jet. The variation of coating removal width measured in the experiments is shown in Figure 6. Using fixed target distance and the same kind of nozzle is to eliminate the influence of target distance and nozzle structure on paint removal width, so that the jet impacting force is a fixed value under a certain inlet pressure. The local impingement time on the micro section of paint removal zone is changed by traversal velocity. In this paper, the coating removal width is the width of the removal area where the coating is completely removed and the interface between coating and steel plate is exposed. When the inlet pressure of the jet is 30 MPa, the traversal velocity exceeds 0.7 mm/s, the single impact of the jet can destroy the coating but not impinge on the contact surface between coating and steel plate, which means the coating is not completely removed. Thus, there is no removal width value. When the jet inlet pressure is 30 MPa, the critical traverse speed of the single traversal coating removal is 0.7 mm/s. When the jet inlet pressure is 28 MPa, the critical traverse speed is 0.4 mm/s. Experiment results show that coating removal width increases with jet pressure. When the traversal speed increases, the value of removal width becomes smaller. With the increase of traversal speed, local exposure time of jet impingement decreases. It is necessary to analyze the effect of local exposure time on removal width.

Coating removal by water jet is a complex physical process, which is mainly affected by jet pressure, target distance, traversal speed, coating thickness, and surface roughness of the steel plate. It is difficult to describe the correlation of these parameters to coating removal rate by formula. Therefore, most researches carry out numerical analysis on experiment results, and obtain empirical formulas under specific experimental conditions. Based on the numerical analysis of the experimental results, the fitting formula of the effect of inlet pressure *P* and traversal speed *v* on the coating removal width *w* is obtained as follows:(1)$$w=-1.041-3.384v+0.007P+0.869v^{2}+0.003P^{2}+0.041v\cdot P$$

The R-square of this fitting model is 0.973. According to this fitting equation, when the jet pressure is 28 MPa, and the traversal speeds are 0.2 mm/s and 0.25 mm/s, the calculation results of depainting width are 1.09 mm and 1.01 mm. And the experimental measurement results are 1.15 mm and 1.05 mm. the error are 5.2% and 3.8% The calculation results of the fitting equation can well fit the experimental data.

### 3.3. Coating Removal by Repeated Jet Impingement

In the actual water jet coating removal operation, there is a method of repeated cleaning to achieve a higher removal rate. In order to study the influence of water jet repeated impact times on paint removal width, multiple impingement test of water jet with fixed traversal speed is designed. Water jet pressure is 32 MPa and 28 MPa, the traversal speed is 1 mm/s, the number of traversal varies from 1 time to 20 times. Water jet repeatedly impacts the same area of the painted plate with the fixed traversal velocity, then the width of coating removal is measured. The measurement results of coating removal width are shown in Figure 7a. The partial enlarged view is shown in Figure 7b. the length of the scale is 2 mm and the division value is 0.05 mm. When the inlet pressure is 28 MPa, the impact of the jet with the traversal speed of 1 mm/s cannot completely destroy the coating and exposure the contact surface of the steel plate, thus there is no measured value. It can be seen from the Figure 8 that with the increase of the number of water jet impingement, the width of paint removal increases at first and then tends to a fixed value. From the point view of energy-based model, the failure of coating adhering on rough surface by jet impingement is that the kinetic energy transferred from micro jet flow to coating increases gradually with exposure time and reaches the threshold of coating failure energy. From a microscopic point of view, the surface roughness reduces the shearing area of the diffusion jet along the wall, so the energy accumulation in the coating mainly comes from the kinetic energy of the velocity component perpendicular to the painted plate. From a macro point of view, the impact force of the jet is greater than the failure strength. The coating is removed from the rough surface by water jet impingement. Therefore, the maximum width of coating removal after repeated cleaning should be the diameter of the jet velocity distribution area. From the simulation results, the maximum width of the paint removal on the rough surface is 2*r*.

### 3.4. Coating Removal by Jet Impact with Fixed Total Exposure Time

The single variable speed paint removal test and the constant speed multiple cleaning test show that the width of coating removal increases with the local exposure time of water jet impingement on the micro segment of the removal area. It is necessary to further analyze the effect of total exposure time, traversal speed and repeated cleaning times on the width of coating removal at constant water jet pressure. Experiment of water jet repeated impingement with constant total exposure time is designed as:

In this experiment, single moving displacement of the jet gun is 180 mm, the total exposure time *t_c_* can be expressed as:(2)$$t_{c}=c\cdot \frac{l}{v}$$
where *c* is the number of repeated cleaning times and *v* is the traversal speed of jet. In test 1, traversal speed varies from 1 mm/s to 5 mm/s, total exposure time is 15 min and 30 min. Repeated cleaning times at different speeds for a constant total exposure time are shown in Table 2, and the values of removal width are shown in Figure 9a.

In test 2, the traversal speed is 1 mm/s and 2 mm/s, the total exposure time ranges from 3 min to 30 min. Table 3. displays cleaning frequency of different total exposure time. The result of test 2 is shown in Figure 9b.

Figure 9 confirms that the width of coating-removal increases with the total exposure time. When the total exposure time is constant, the value of coating removal width is fixed, independent with traversal speed and repeated times. That is:(3)$$d_{1}=f(v_{1},n_{1})$$
(4)$$d_{2}=f(v_{2},n_{2})$$
(5)$$t_{1}=\frac{L}{v_{1}}\cdot n_{1}$$
(6)$$t_{2}=\frac{L}{v_{2}}\cdot n_{2}$$
where *f(v, n)* is a function of traversal speed *v* and the number of repeated impingement times *n*. When *t*_1_ = *t*_2_, that is $L\cdot n_{1}/v_{1}=L\cdot n_{2}/v_{2}$, *d*_1_ is equal to *d*_2_.

## 4. Characteristics of Coating Removal by Water Jet Rotating Cleaning Disc

In the actual jet paint removal and cleaning operation, the water jet rotating cleaning disc is used for large area plate cleaning. The physical and structure of the device are shown in Figure 10. The nozzle is installed in the rotating rod inside the disc. During paint removal operation, the pneumatic motor in the cleaning plate drives the rotating rod with high rotational speed, and the cleaning plate moves to achieve coating removal.

The nozzle in the disc chamber is moving with a resultant motion form that consists of a circular motion with radius *R* and angular velocity *ω* and a linear motion with moving speed *v*. The trajectory can be described as a cycloid. Its parametric equations are expressed by the following formulas:(7)$$\left\{ \begin{array}{l} x_{1}=R\cos\omega t+vt\\ y_{1}=R\sin\omega t \end{array}\right.$$

The trajectory of the nozzle axis is shown in Figure 11, and the shape is a cycloid. The trajectory has a self-intersection point when $v\cdot \frac{2\pi }{\omega }\leq 2R$. The height interval of each intersection is different, resulting in the sparse distribution of trajectory lines near the *x*-axis and dense distribution near the upper and lower boundary of the trajectory. The paint removal trajectory coverage diagram is shown in Figure 12. Too fast moving speed *v* causes the trajectory interval *H* to be larger than the coating removal width *w* of a single jet, so that the coating removal zone cannot completely cover the traversal area. The height difference between the intersection points near the upper and lower boundary of the trajectory is small with the paint removal trajectory densely distributed. The region near the boundary of trajectory is completely cleaned because of the overplus coverage of coating removal area. If the width of the residual coating is *d*, the width of the clean area near the edge of the cleaning plate is *R*-*d*/2, and the maximum interval of the trajectory at the intersection point of the *x* axis is *H*_max_. If the rotating cleaning disc can completely remove the coating in this area at one time, that is, the width of the residual coating *d* = 0, it is necessary to reduce the moving speed *v* and make the maximum trajectory interval $H_{\max}\leq w$. When *H*_max_ is equal to *w*, the maximum moving speed of removing the coating once is *v*_max_.

According to the trajectory analysis results, the experiment of rotating cleaning disc coating removal with different moving speed is carried out. The average rotational speed is 3000 r/min, and the moving speed is 1.2 m/min, 0.9 m/min, and 0.6 m/min. Coating residue on the steel plate is shown in Figure 13. Result shows that there is a clear removal zone near the boundary of the trajectory when the cleaning plate device performs the jet coating removal operation with high moving speed. There are some bright spots in the coating residual region. These spots are bulges on the rough steel plate where the jet target distance of these spots is smaller than the constant value. Therefore, the kinetic energy at these spots is higher and the removal effect is better compared with other regions. The test result is consistent with the results of simulation and theoretical analysis.

## 5. Conclusions

Based on the experimental and simulation methods, this paper analyzes the area of water jet impingement and the effect of exposure time on the width of micro jet flow coating removal on rough surface. According to the experiment results, the efficiency and parameter optimization of the rotating disc coating removal are analyzed. The influence of the cleaning disc moving speed on the cleaning rate of the coating is studied by analyzing the movement of the micro jet flow and the trajectory distribution of the paint removal. Experiment and the calculation results show that:

(1) The simulation result shows that the inlet water pressure and the micro flow channel of nozzle mainly affect the jet outlet flow velocity. Jet velocity attenuation and increment of impingement area result from increasing target distance. The micro flow velocity has the characteristics that the maximal velocity occurred in the axial region and the velocity attenuated along the radial direction.

(2) Reducing the traverse speed or increasing the number of repeated impingement times can increase the local exposure time of the jet on the trajectory micro segment, which can improve the coating removal width of the jet.

(3) The results of a single-beam moving jet coating removal test show that when the jet pressure is constant, the paint removal width is positively correlated with the total exposure time. When the total exposure time is constant, the coating removal width has a fixed value, which is not affected by the traversal speed or the number of traversals.

(4) The moving speed of the rotating cleaning disc mainly affects the width of the residual coating on the surface after the coating removal. With the increase of the moving speed v, the trajectory is sparser and the removal strips cannot fully cover the traversal area, which causes coating residual. It is necessary to slow down the moving speed v, so as to totally remove the coating by once traversal. This study is helpful to optimize the jet cleaning parameters.

## Figures and Tables

**Figure 1 micromachines-12-00173-f001:**
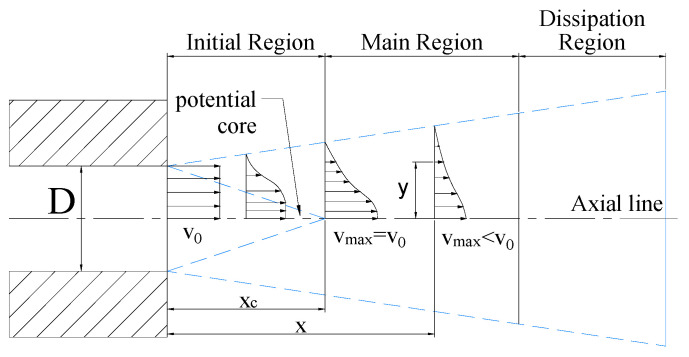
Structure and fluid velocity distribution of water jet, where *D* is diameter of micro flow channel in the nozzle, *v*_0_ is velocity at the nozzle outlet, *x_c_* is the length of the potential core, *x* is the target distance, *v_max_* is the velocity on the axial line at different target distance, *y* is the distance from a point to the axis on the cross section of a certain target distance.

**Figure 2 micromachines-12-00173-f002:**
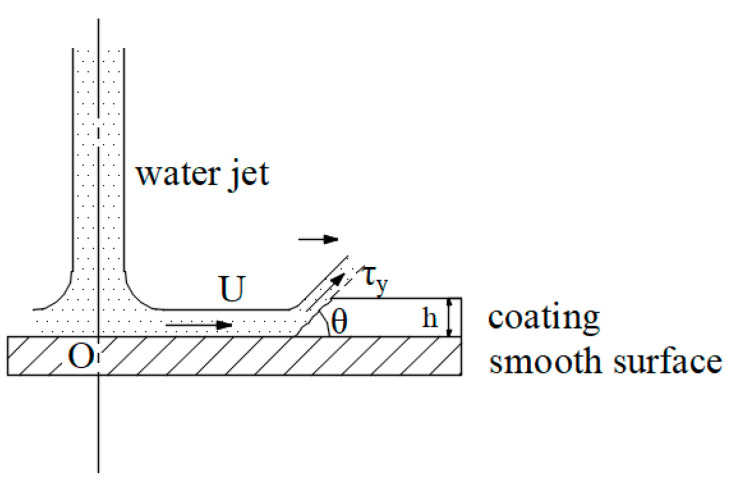
Principle of micro jet flow coating removal on the smooth surface, where *U* is fluid velocity of diffusing jet, *θ* is angle of residual coating surface, *τ_y_* is the shear yield stress of the layer, *h* is thickness of coating.

**Figure 3 micromachines-12-00173-f003:**
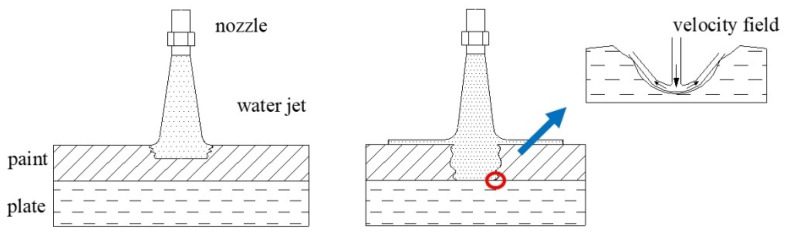
Schematic of micro jet flow coating removal on the roughness surface.

**Figure 4 micromachines-12-00173-f004:**
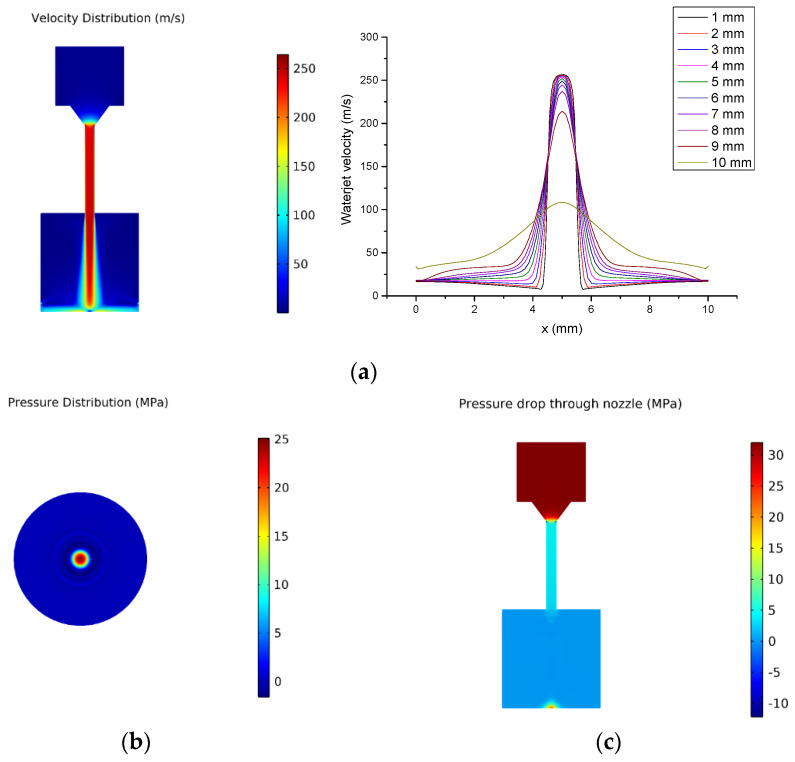
Simulation of water jet impinging on the wall: (**a**) Velocity distribution of water jet; (**b**) Impact pressure distribution of water jet; (**c**) Pressure drop through the nozzle.

**Figure 5 micromachines-12-00173-f005:**
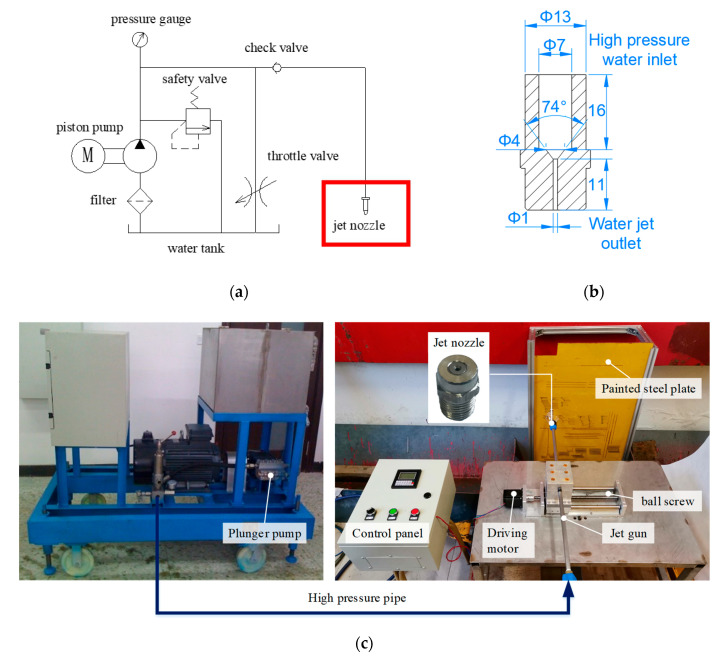
High pressure water jet paint removal experimental system: (**a**) The principle of water jet generating; (**b**) Structure size of nozzle; (**c**) Coating removing test bench with water jet. High pressure water pressurized by the piston pump is sent to the jet gun through the pipeline. When the jet gun starts straight reciprocating motion driven by the slipway, the whole system begins the paint removal test.

**Figure 6 micromachines-12-00173-f006:**
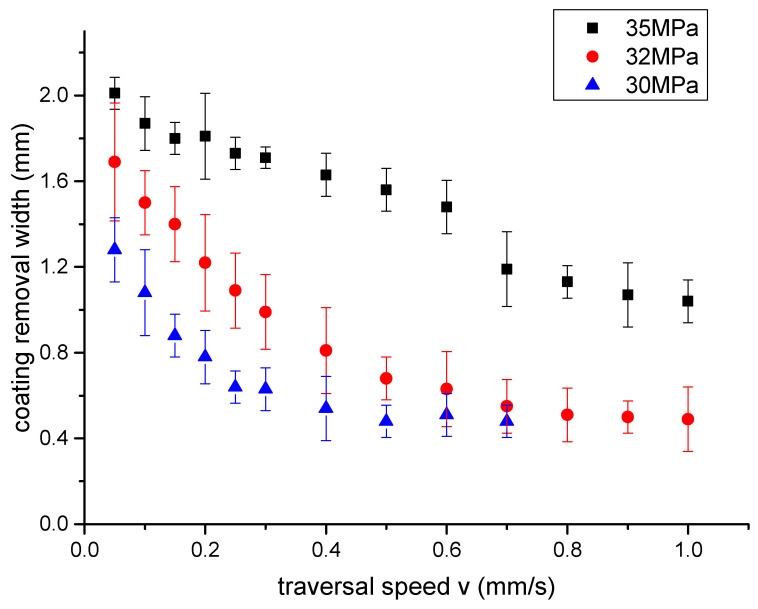
Effect of the moving speed on the width of coating removal. The width gradually reduces as the velocity increases.

**Figure 7 micromachines-12-00173-f007:**
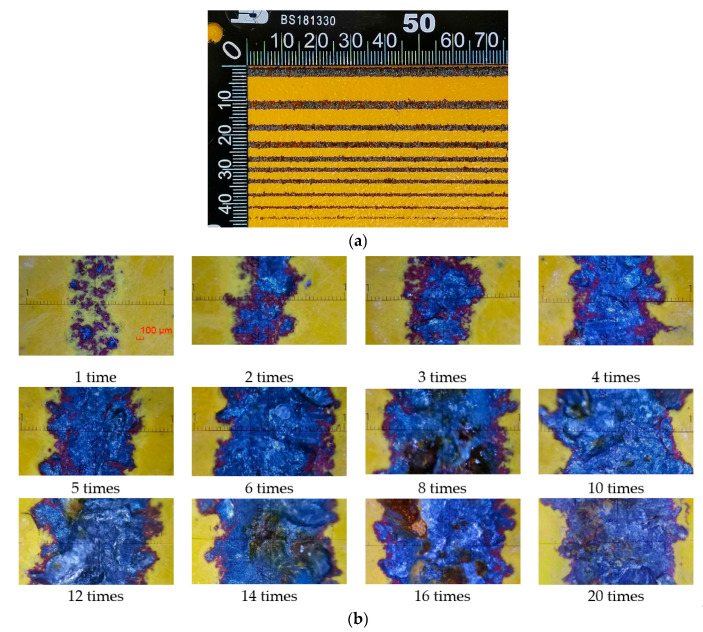
Comparison of removal effects with different traversal times: (**a**) Coating removal effects by water jet with different traversal times; (**b**) Enlarged images of coating removal effects.

**Figure 8 micromachines-12-00173-f008:**
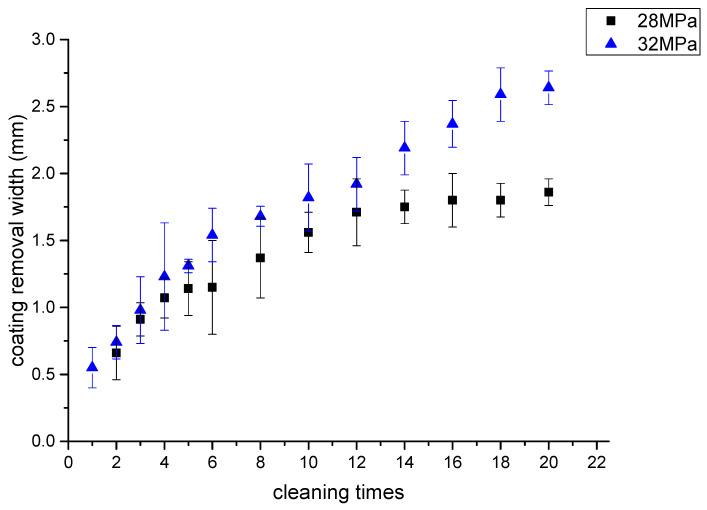
Effect of jet impact times on coating removal width.

**Figure 9 micromachines-12-00173-f009:**
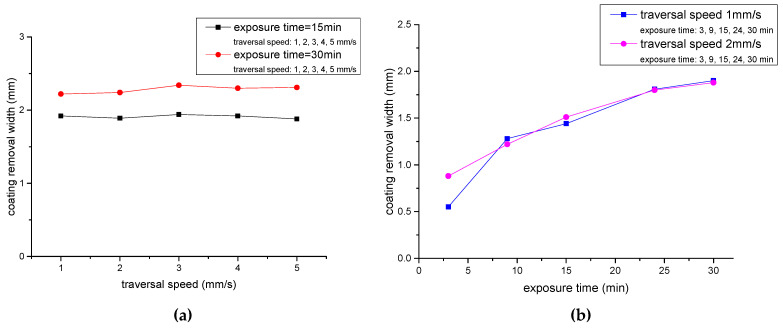
Multiple coating removal by waterjet with fixed total exposure time: (**a**) Coating removal width is independent with traversal speed and repeated impingement times when the total exposure time is fixed. (**b**) Removal width increases with total exposure time.

**Figure 10 micromachines-12-00173-f010:**
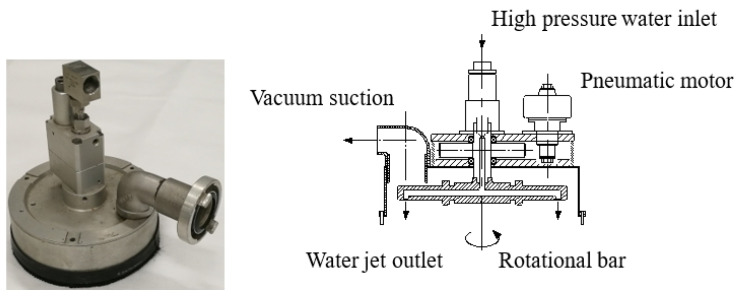
Structure of water jet rotating cleaning disc.

**Figure 11 micromachines-12-00173-f011:**
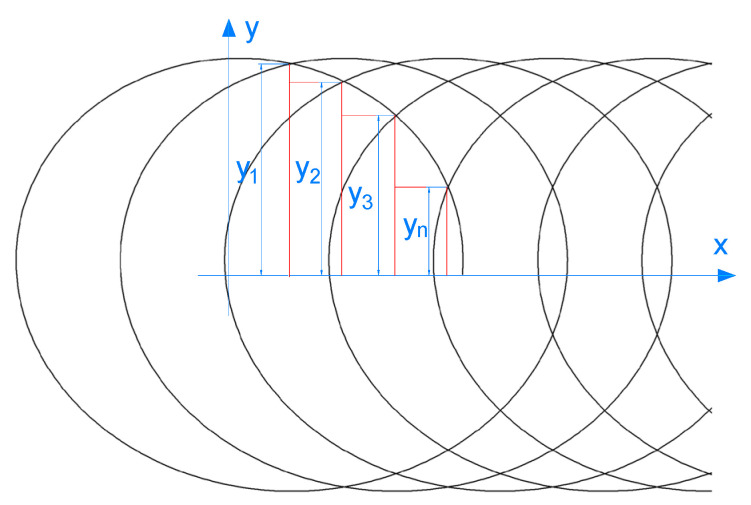
Trajectory of nozzle in the disc chamber.

**Figure 12 micromachines-12-00173-f012:**
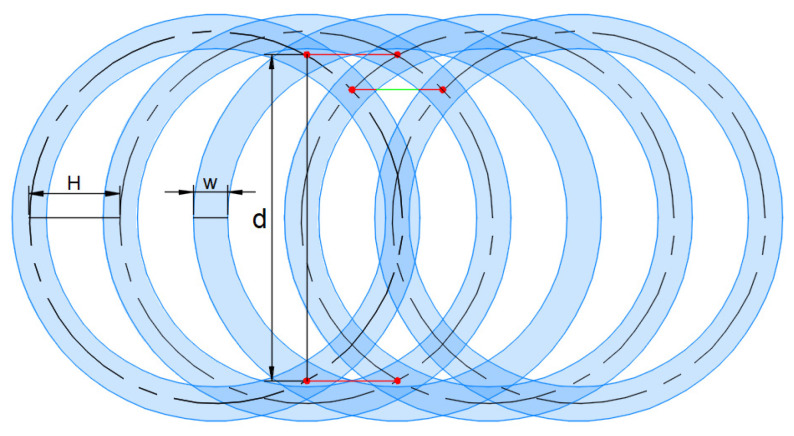
Schematic diagram of residual coating with high moving speed.

**Figure 13 micromachines-12-00173-f013:**
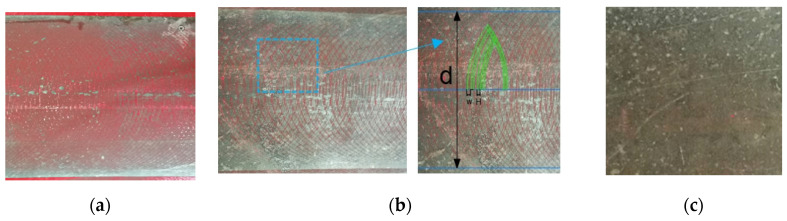
Surface effect of coating removal by cleaning disc with various moving speed: (**a**) 180 MPa, 1.2 m/min; (**b**) 0.9 m/min; (**c**) 0.6 m/min.

**Table 1 micromachines-12-00173-t001:** Parameters of various water jet equipment.

Car Washing	Coal Mining	Cleaning for Industry	Cutting and Crushing	Machining	Water Jet Canon
7 MPa 20 L/min	7 MPa 4000 L/min	14~300 MPa 10~80 L/min	70~100 MPa 40~200 L/min	200~400 MPa 4 L/min	20~700 MPa 40~80 L/min

**Table 2 micromachines-12-00173-t002:** Relationship between cleaning times and traversal speed during a constant exposure time.

No.	*v* (mm/s)	Total Time (min)	Cleaning Times
1	1	15	5
2	2	15	10
3	3	15	15
4	4	15	20
5	5	15	25
6	1	30	10
7	2	30	20
8	3	30	30
9	4	30	40
10	5	30	50

**Table 3 micromachines-12-00173-t003:** Relationship between cleaning times and traversal speed in different total exposure time.

No.	*v* (mm/s)	Total Time (min)	Cleaning Times
1	1	3	1
2	1	9	3
3	1	15	5
4	1	24	8
5	1	30	10
6	2	3	2
7	2	9	6
8	2	15	10
9	2	24	16
10	2	30	20

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
