# Peer review of "Experimental Study on the Coating Removing Characteristics of High-Pressure Water Jet by Micro Jet Flow"

_micromachines, 2021, doi:10.3390/mi12020173_

Round 1

Reviewer 1 Report

The paper analyzes the area of water jet impingement and the effect of exposure time on the width of micro jet flow coating removal on rough surface. The efficiency and parameter optimization of the rotating disc coating removal are analyzed.

An article at an average scientific level, but of great importance for industrial practice. Editorial notes:

  1. illegible markings of regions in fig. 1
  2. describe the symbols in fig. 2
  3. illegible markings in fig. 5b

Author Response

Point 1: Illegible markings of regions in fig. 1. 

Response 1: Thank you for your helpful comment. We have change the markings in figure 1, which was marked as region â… , â…¡ and â…¢ before. (Page 4, line 160)

Point 2: Describe the symbols in fig. 2.

Response 2: Your comment is helpful to describe the coating damage model. We have describe the markings in figure 2 in red. (Page 4, line 174-176)

Point 3: Illegible markings in fig. 5b.

Response 3: Thank you for your comment. We have marked the inlet of high pressure water and outlet of water jet in figure 5b. (Page 6, line 222)

Reviewer 2 Report

The paper mainly discusses numerical and experiment tests for high-speed water jet for paint clearing. The review suggests that the manuscript needs to be revised before publication.

  1. It will be great if any experiment validation of the numerical model is included
  2. In the model, it may be useful to address the correlation between impact pressure and exposure time to removal rate, or the critical pressure to remove the coating.
  3. Pressure drop through the nozzle would be useful information to include as it determines what kind of pumping system is needed.

Author Response

Point 1: It will be great if any experiment validation of the numerical model is included. 

Response 1: Thank you for your helpful suggestion, we will analyse the effects of other parameters on coating removal width and try to establish a numerical model in further research. Coating removal by water jet is a complex physical process, which is mainly affected by jet pressure, target distance, traversal speed, coating thickness, surface roughness of steel plate and other parameters. It is difficult to describe the correlation of these parame-ters to coating removal rate by formula. Therefore, most researches carry out numerical analysis on experiment results, and obtain empirical formulas under specific experimental conditions. So we obtain a fitting equation and verified by experimental results. (Page 7, line 251-264)

Point 2: In the model, it may be useful to address the correlation between impact pressure and exposure time to removal rate, or the critical pressure to remove the coating.

Response 2: Your comments is thoughtful. We take numerical analysis on experimental results and get a fitting equation in order to show the correlation between inlet pressure and traversal speed to the coating removal width.(Page 7, line251-264) The critical traversal speed of different inlet pressure is also shown in the paper according to the experiment results.(Page 7, line 244-246)

Point 3: Pressure drop through the nozzle would be useful information to include as it determines what kind of pumping system is needed.

Response 3: Fig 4c is added in this paper. (Page 5, line 195-198, Page 6, line 206) Fig 4c is the simulation result of the pressure drop through the nozzle caused by the energy conversion. High pressure water at the inlet has a large pressure energy. After flowing through the micro fluid channel, the pressure energy of the water converts into kinetic energy to form a high-speed jet.
